# Geographic Distribution, Phenotype and Epidemiological Tendency in Inflammatory Bowel Disease Patients in Romania

**DOI:** 10.3390/medicina55100704

**Published:** 2019-10-20

**Authors:** Adrian Goldiș, Raluca Lupușoru, Liana Gheorghe, Cristian Gheorghe, Anca Trifan, Daniela Dobru, Cristina Cijevschi, Alina Tanțău, Gabriel Constantinescu, Răzvan Iacob, Ramona Goldiș, Mircea Diculescu

**Affiliations:** 1Department of Gastroenterology and Hepatology, “Victor Babeș” University of Medicine and Pharmacy, 300041 Timișoara, Romania; goldisadi@yahoo.com; 2Department of Functional Sciences, “Victor Babeș” University of Medicine and Pharmacy, 300041 Timișoara, Romania; 3Department of Gastroenterology and Hepatology, “Carol Davila” University of Medicine and Pharmacy, Fundeni Clinical Institute, 020021 Bucharest, Romania; drlgheorghe@gmail.com (L.G.); drgheorghe@xnet.ro (C.G.); raziacob@gmail.com (R.I.); mmdiculescu@yahoo.com (M.D.); 4Gastroenterology and Hepatology Institute, “Grigore T. Popa” University of Medicine and Pharmacy, 700019 Iași, Romania; ancatrifan@yahoo.com (A.T.); cristinacijevschi@yahoo.com (C.C.); 5Municipal Hospital, Gastroenterology, University of Medicine and Pharmacy, 540139 Târgu-Mureș, Romania; danidobru@gmail.com; 6“Iuliu Hațieganu” University of Medicine and Pharmacy, 3rd Medical Clinic, 400012 Cluj-Napoca, Romania; alina.i.tantau@gmail.com; 7Department of Gastroenterology and Hepatology, “Carol Davila” University of Medicine and Pharmacy, Floreasca Emergency Hospital, 020021 Bucharest, Romania; gabrielconstantinescu63@gmail.com; 8Algomed Policlinic, 300041 Timisoara, Romania; amalia_goldis@yahoo.com

**Keywords:** inflammatory bowel disease, ulcerative colitis, Crohn’s disease, epidemiology, phenotypes, IBD treatment

## Abstract

*Background and objective:* The incidence of inflammatory bowel disease (IBD) over the past years in Romania has been on the rise, but epidemiologic data are lacking. The aim of this study was to define the characteristics of IBD, the trends and phenotype among IBD patients in Romania. *Material and methods:* We conducted a prospective study over a period of 12 years, from 2006 to 2017. All patients diagnosed with IBD on clinical, radiological, endoscopic and histological features were included. We divided the country into eight regions: west (W), north-east (NE), north-west (NW), south-east (SE), south-west (SW), south (S), central (C) and Bucharest-Ilfov (B), and data were analyzed accordingly. *Results:* A total of 2724 patients were included in this database, but only 2248 were included in the final analysis, with all data available. Of the 2248 patients, 935 were Crohn’s disease (CD), 1263 were ulcerative colitis (UC) and 50 were IBD-undetermined. In UC phenotypes we observed more frequent left-sided colitis (50.5%, *p* < 0.0001), and in CD phenotype we observed more frequent colonic and ileo-colonic localization (37.8% and 37.6%, *p* < 0.0001). The region with the most IBD cases was NE (25.1%) and with the least IBD cases was SW (4.9%). UC was found more frequently in NE (32%), while CD was found more frequently in Bucharest (28.6%). *Conclusions:* In Romania, ulcerative colitis is more frequent than CD. UC is predominant in the northern part of Romania, while CD has become predominant in the southern part of the country. IBD occurs more in the male population, and in urban and industrialized areas. There are differences between the regions in Romania regarding IBD phenotypes, gender distributions, age distribution, treatment, smoking status and complications.

## 1. Introduction

Inflammatory bowel disease (IBD) is a chronic, relapsing, inflammatory disorder of the gastrointestinal tract and includes ulcerative colitis (UC) and Crohn’s disease (CD), which shows differences in the pathology and clinical characteristics. Currently, the etiology and pathogenesis of IBD are still poorly understood. It is widely accepted that the pathogenesis of IBD comprises genetic factors and environmental factors [1]. More than 100 genes have been identified by the genome-wide association scan to increase the susceptibility to IBD [2]. However, genetic susceptibility cannot completely explain the high incidence and prevalence of IBD observed in developed and developing countries [3].

Information regarding the epidemiology of IBD world-wide is poor, especially information about Romanian IBD patients. In an IBD review, the prevalence of ulcerative colitis was described as 249 per 100,000 persons for North America and 505 per 100,000 persons for Europe; the incidence for ulcerative colitis was 19.2 per 100,000 persons for North America and 24.3 per 100,000 persons for Europe, 6.3 per 100,000 persons in Asia and the Middle East. For Crohn’s disease, the prevalence was 319 per 100,000 persons in North America and 322 per 100,000 persons in Europe. The annual incidence was 20.2 per 100,000 persons in North America, 12.7 per 100,000 persons in Europe and 5 per 100,000 persons in Asia and Middle East [4].

The purpose of this study was to evaluate the epidemiology, the trends, phenotypes and treatment among patients with IBD in Romania. 

## 2. Materials and Methods

### 2.1. Study Design

The “IBDPROSPECT” database was conducted in 10 Romanian University Centers. It is a national web-accessible database that started in 2006 through the partnership of two medical centers. The IBDPROSPECT subsequently came under the remit of the Romanian Crohn’s and Colitis Club, a non-profit professional organization established as a professional and scientific body of the Romanian Society of Gastroenterology and Hepatology which was interested in deepening the fundamental knowledge, diagnosis and treatment of IBD in Romania. The database can generate important insights into the dynamics of IBD in our country. The database allows the generation of new sets of specific variables for future projects that would have as their starting points the previously recorded data sets [5].

We included all patients diagnosed with IBD based on clinical, radiological, endoscopic and histological features. The population was demographically similar in all centers. Variables collected included age, gender, date of diagnosis, family history, smoking status, complication, phenotype, endoscopic, imaging, laboratory data and demographic data were annotated. 

The diagnosis of IBD and the phenotype were established according to the Montreal criteria [4] for Crohn’s disease (CD) and ulcerative colitis (UC). For patients whose endoscopic examination, image and histopathological examination and laboratorial results were associated with medical reports describing the difficulty of diagnosing either CD or UC, the terminology ‘unclassified inflammatory bowel disease’ (UIBD) was applied.

For the final analysis, we divided the country into eight regions (Figure 1): west (W), north-east (NE), north-west (NW), south-east (SE), south-west (SW), south (S), central (C) and Bucharest-Ilfov (B), to see if there were any differences between the regions regarding the epidemiology. The west region included the following counties: Timis (TM), Arad (AR), Caras-Severin (CS) and Hunedoara (HD). The north-east region included: Bacau (BC), Botosani (BT), Iasi (IS), Neamt (NT), Suceava (SV) and Vaslui (VS). The south-east region included: Braila (BR), Buzau (BZ), Constanta (CT), Galati (GL), Tulcea (TL) and Vrancea (VN). The south region included: Arges (AG), Calarasi (CL), Dambovita (DB), Giurgiu (GR), Ialomita (IL), Prahova (PH) and Teleorman (TL). The Central region included: Mures (MS), Harghita (HR), Covasna (CV), Brasov (BV), Sibiu (SB) and Alba (AB). The south-west region included: Dolj (DJ), Gorj (GJ), Mehedinti (MH), Olt (OT) and Valcea (VL). The north-west region included: Bihor (BH), Bistrita (BN), Cluj (CJ), Maramures (MM), Satu-Mare (SM) and Salaj (SJ). 

### 2.2. Sample Analysed

The sample included 2724 patients with IBD in Romania which were recorded in IBDPROSPECT from January 2006 to June 2017. Treatment and phenotype were not recorded for 476 patients, so data for 2248 patients were used. 

No Ethical approval was needed to conduct this study, as it is a noninterventional retrospective study, therefore according to Romanian legislation, no approval was needed. Furthermore, the patients from whom the samples were collected have all signed written informed consent according to the Helsinki Declaration.

### 2.3. Statistical Analysis

The Kolmogorov–Smirnov test was used for testing the distribution of numerical variables. Qualitative variables were presented as numbers and percentages. Parametric tests (*t*-test, ANOVA) were used for the assessment of differences between numerical variables with normal distribution; and nonparametric tests (Mann–Whitney or Kruskal–Wallis tests) for variables with non-normal distribution. The Chi-square (χ^2^) test was used for comparing proportions expressed as percentages (“*n*” designates the total number of patients included in a particular subgroup). Furthermore, 95% confidence intervals were calculated for each predictive test and a *p*-value < 0.05 was considered significant for all statistical tests. The statistical analysis was performed using SPSS software, Version 20.0 (IBM SPSS Statistics) and Microsoft Office Excel 2019. 

## 3. Results

Our IBD prospective database was comprised of 2724 patients. The final analysis included 2248 patients with all data available; 1200 were male and 1048 were female. The mean age was 39.48 ± 15.82. Patients characteristics are presented in Table 1. 

We used the eight regions division, so the distribution of the IBD between them is in Table 2 and the IBD distribution by phenotype in Figure 2. The region with the most IBD cases was north-east and the region with the lowest IBD cases was south-west, *p* < 0.001. A peak was evidenced in IBD between the age of 31–40 and in the lowest cases between ages 0–10, *p* < 0.0001. (Figure 3).

As it was so insignificant, we excluded UIBD from the final analysis, to eliminate bias. 

When we compared the two IBD groups (Table 2), we found out that there were differences between the groups regarding age, gender, provenance, treatment and smoking conditions.

The phenotypes evaluated are presented in Table 3 and Table 4. For UC, the group of pediatric patients (aged under 16 years old) was the group with the lowest number of patients. In the age group 17–40 years, left-sided colitis was more frequent (*p* < 0.0001), and in the age group of more than 40 years, left-sided colitis was also more frequent (*p* < 0.0001). Pancolitis was more frequent in age group >40, *p* = 0.03. In CD, colonic localization and nonstricturing, nonpenetration behavior was more frequent (Table 3).

When we compared the IBD patients between the eight regions of Romania we found out that there were significant differences between them (*p* < 0.05), regarding gender distribution, number of cases, IBD distribution and complications. The region with the most cases was NE and the region with the less IBD cases was SW. Regarding the phenotypes, the predominant UC patients were found in NE, NW and C. The predominant CD patients were found in B, W, S and SE regions. In the south, the proportion of ulcerative colitis and Crohn’s disease was equal (Figure 4).

To understand why there are differences between phenotypes and regions, we conducted a univariate analysis (Table 5) to see if there were any specific factors involved. For the north region with the ulcerative colitis predominance, we found out that age above 40 years, male gender, positive family history and rural provenance were associated with the presence of ulcerative colitis in that area. For the south region with the predominance of Crohn’s disease, age under 40 years, male gender and smoking were associated with the presence of Crohn’s disease. In the south-west region, phenotypes distribution demonstrated equality while the factors involved were being aged above 40 years. 

We divided the patients into eight regions, but also according to phenotypes, to establish what were the differences. Regarding the age distribution, smoking, treatment and complications, we found a significant statistical difference, *p* < 0.05. 

The tendency in diagnosis has increased significantly and gradually over the years, from 9 in the first years to 631 in the last 5 years, and a peak between 2008–2012, *p* < 0.0001, but we stopped the cases’ selection in the middle of 2017 (Figure 5). The rise in numbers seems to begin in 1988, *p* < 0.0001.

## 4. Discussion

In this comprehensive epidemiological evaluation of IBD in Romania, we observed an increase of diagnosed IBD patients over the years. Starting with year 1988, many more people have been diagnosed with IBD. This can be explained by the fact that people nowadays have easy access to a doctor, and because access to information is much more available. We could not evaluate the incidence of IBD in the Romanian population because in this database only patients from University centers were included. In Romania, the private sector of health is very developed, and sometimes it is easier for patients to go private as they have direct access to the doctor, and have access to the same treatment offered by the state. 

This study contributes to the knowledge of the epidemiology of IBD in Romania. It provides data about the regional distribution and regional differences in IBD in Romania. 

The overview of this study concludes that UC is more frequent in Romania, similar to other studies [7,8,9,10,11]. The UC to CD ratio was 1.35:1.

The gender distribution was similar to previous studies [4]. The male to female ratio was 1.14:1. IBD was more frequent in urban areas. This fact can be explained because the urban population is more stressed, patients do not have a proper alimentary habit, or simply because the rural people do not have the same access to medical services [12,13]. As is described in other studies, the group age 31–40 years was with the most IBD cases, no matter the phenotype [14]. In pediatric patients (<17 years), 24 had IBD, among these 14/24 had UC. A similar frequency was found in studies reported in the literature [15,16,17]. In Romania, the trend of age diagnosis seems to be similar to studies reported in literature: CD was diagnosed at a younger age than UC. Regarding treatment, we found that biologicals are given more often in patients with CD, a fact that can suggest severe forms of CD. Aminosalicylates had a higher frequency also in CD patients, which is contrary to the guidelines, but probably because of the frequent colonic and ileocolonic localization of CD [18,19,20,21]. Among patients with UC, left-sided colitis was the most frequent, similar to other studies [14,22]. For CD patients, nonstricturing, nonpenetration behavior was the most frequent, in contrast to previous studies [23,24].

Our country is divided into eight regions. The north-east region seems to be with a higher proportion of IBD patients, especially UC patients. The higher proportion of CD patients was found in the Bucharest region. If we divided the country into two regions, north (NE, NW and C) and south (W, SV, B and SE) we can tell that the north region has the majority of UC patients, while the south region has the majority of CD patients. It is known that CD occurs in more industrialized regions, so Romania follows the same pattern because the south region is very industrialized [25,26,27,28]. Comparing the eight regions by phenotype for UC we found differences in gender distribution, age, family history and treatment. For CD we found differences in gender distribution, age, smoking status, biological treatment and complications. The factors involved in the presence of the predominant phenotype in regions was: for the north region with UC predominance, we found out that age above 40 years, male gender, positive family history and rural provenance were associated with the presence of ulcerative colitis in that area. For the south region with the predominance of CD, age under 40 years, male gender, smoking and lower gastrointestinal bleeding was associated with the presence of CD. In the south-west region, there was equality regarding phenotype distribution and the factors involved in this were the age above 40 years old. 

The study had some limitations; the patients were inscribed only from university centers so it is not entirely a population-based study, and not all the medical doctors from these university centers contributed to this study, so further studies are required. A strength for this study may be the large number of the cohort population, which on a “sampling size” analysis is higher than the number onset for the biggest test power, so the results are representative. Another strength of the study is that it gives information regarding phenotypes, trends and medication in IBD in Romania. 

## 5. Conclusions

In Romania, UC is more frequent than CD. UC is predominant in the northern part of Romania, while CD has become predominant in the southern part of the country. IBD occurs more in male population, urban and industrialized areas. There are differences between the regions in Romania regarding IBD phenotypes, gender distributions, age distribution, treatment, smoking status and complications.

## 6. Patents

This article was submitted on behalf of the IBDPROSPECT Study Group.

## Figures and Tables

**Figure 1 medicina-55-00704-f001:**
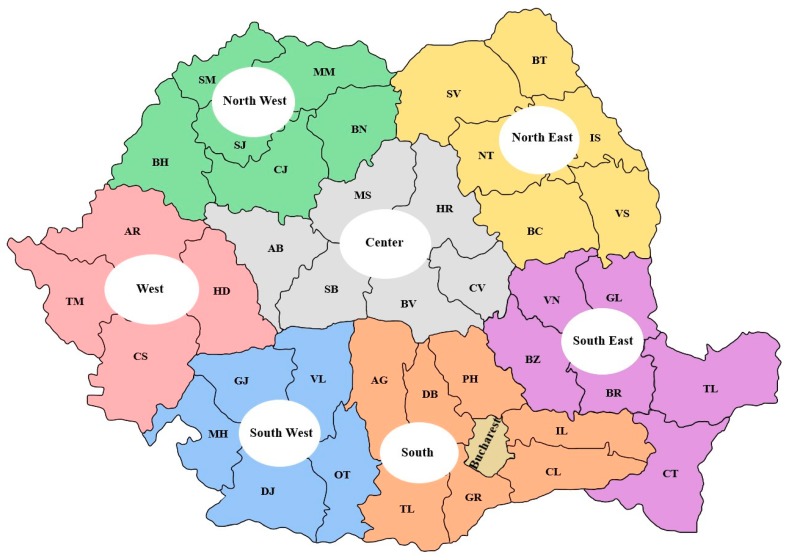
Romania-geographical regions [6].

**Figure 2 medicina-55-00704-f002:**
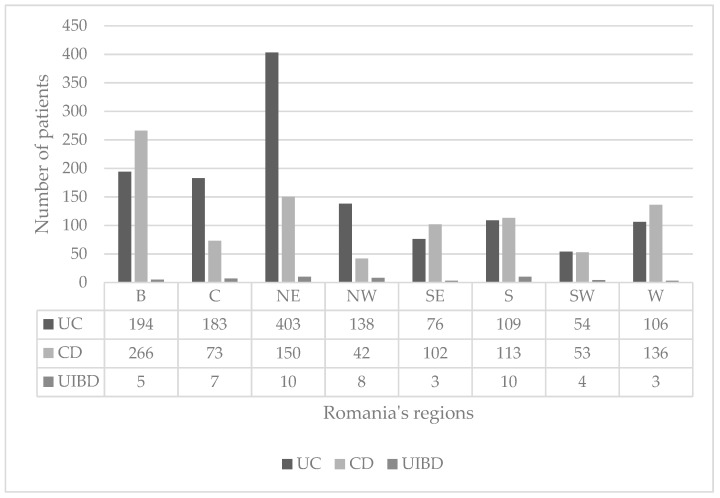
Inflammatory bowel disease (IBD) phenotype distribution over the eight regions. (B = Bucharest, C = Center, NE = North East, NW = North West, SE = South East, S = South, SW = South West, W = West; UC = Ulcerative Colitis, CD = Crohn’s Disease, UIBD = unclassified inflammatory bowel disease).

**Figure 3 medicina-55-00704-f003:**
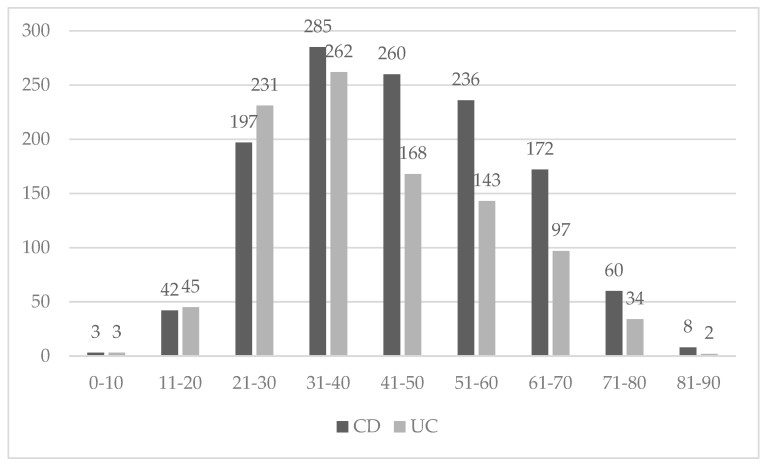
Age group distribution of UC and CD.

**Figure 4 medicina-55-00704-f004:**
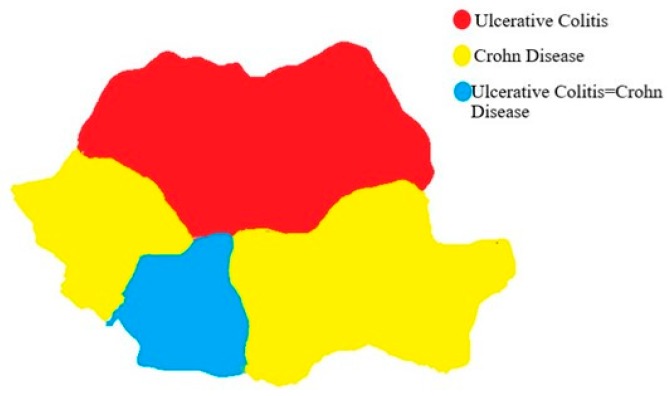
Romania colored map according to IBD phenotypes.

**Figure 5 medicina-55-00704-f005:**
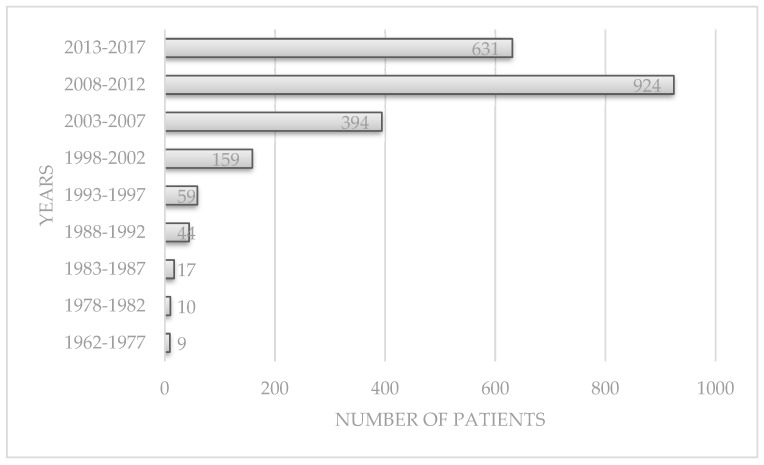
IBD diagnosis over the years.

**Table 1 medicina-55-00704-t001:** Characteristics of the patients.

	*n* (%)/Mean ± SD
Age (years)	39.48 ± 15.82
Gender	
-Female	1048 (46.6%)
-Male	1200 (53.3%)
IBD	
-UC	1263 (56.1%)
-CD	935 (41.5%)
-UIBD	50 (2.4%)
Smoking status	
-Non-smoker	1346 (59.9%)
-Smoker	353 (15.7%)
-Ex-smoker	549 (24.4%)
Provenience	
-Rural	561 (24.9%)
-Urban	1687 (75.1%)
Family history	
-IBD positive	61 (2.7%)
-IBD negative	2187 (97.3%)
Complications	
Abscess	61 (2.7%)
Fistulas	130 (5.7%)
Stenosis	197 (8.7%)
Perforations	17 (0.7%)
Lower GI haemorrhage	329 (14.6%)
Arthritis	175 (7.7%)
Sacroiliitis	21 (0.9%)
Uveitis	29 (1.2%)
Nodular Erythema	0.8%)

*n* = number of patients; SD = standard deviation, IBD = Inflammatory bowel disease, UC = ulcerative colitis, CD = Crohn’s disease, UIBD = unclassified inflammatory bowel disease.

**Table 2 medicina-55-00704-t002:** Comparison of patient characteristics among the IBD groups.

Demographic Variables	IBD		
	*n* = 2248		
	CD	UC	*p*-Value
	*n* (%), Mean ± SD	N (%), Mean ± SD	
Age (years)	41.34 ± 15	45.32 ± 15.47	<0.0001
Gender			
Female	453 (48.4%)	540 (42.7%)	0.009
Male	482 (51.6%)	723 (57.3%)	0.009
Provenance			
Rural	221 (22.5%)	370 (29.3%)	0.0004
Urban	714 (77.5%)	893 (70.7%)	0.0004
Region			
NE	150 (16%)	403 (32%)	<0.0001
NW	42 (4.4%)	138 (11%)	<0.0001
SE	102 (11%)	76 (6%)	<0.0001
SW	53 (5.6%)	54 (4.2%)	0.15
S	113 (12%)	109 (8.6%)	0.01
W	136 (14.5%)	106 (8.3%)	<0.0001
C	73 (7.8%)	183 (14.4%)	<0.0001
B	266 (28.6%)	194 (14.5%)	<0.0001
Treatment			
None	38 (4%)	103 (8.1%)	0.0001
5 ASA	705 (75.4%)	952 (75%)	0.83
Biologicals	70 (7.4%)	100 (7.8%)	0.72
Azathioprine	212 (22.6%)	272 (21.4%)	0.50
Methotrexate	3 (0.3%)	6 (0.4%)	0.69
Combination therapy	25 (2.6%)	11 (0.8%)	0.0008
Corticotrophins	394 (42.1%)	528 (41.6%)	0.81
Smoking status			
Non-smoker	543 (58%)	772 (61.2%)	0.14
Smoker	207 (22.1%)	137 (10.8%)	<0.0001
Ex-smoker	185 (19.9%)	354 (28%)	<0.0001
Family history			
IBD positive	28 (3%)	31 (2.4%)	0.46
IBD negative	907 (97%)	1232 (97.6%)	0.46

SD = standard deviation, n = number of patients, CD = Crohn’s disease, UC = Ulcerative colitis, B = Bucharest, C = Center, NE = North East, NW = North West, SE = South East, S = South, SW = South West, W = West, 5 ASA = 5-aminosalicylic acid (mesalamine).

**Table 3 medicina-55-00704-t003:** Clinical characteristics of ulcerative colitis.

	Extension			
	E1 Proctitis	E2 Left-Sided Colitis	E3 Pancolitis	*p*-Value
Age at diagnosis				
A1 ≤ 16	2/14 (14.2%)	5/14 (35.8%)	7/14 (50%)	0.53
A2 17–40	90/481 (18.7%)	219/481 (47.6%)	172/481 (33.7%)	<0.0001
A3 > 40	135/768 (17.5%)	415/768 (54%)	218/768 (28.5%)	<0.0001
Gender				
Female	102/540 (18.8%)	278/540 (51.4%)	160/540 (29.8%)	<0.0001
Male	106/723 (14.6%)	361/723 (50%)	256/723 (35.4%)	<0.0001

**Table 4 medicina-55-00704-t004:** Clinical characteristics of Crohn’s disease.

		N (%)
Location	L1: terminal ileum	174 (18.6%)
	L2: colonic	354 (37.8%)
	L3 ileocolonic	352 (37.6%)
	L4 isolated upper disease	11 (1.17%)
	L1 + L4	12 (1.28%)
	L3 + L4	15 (1.6%)
Behaviour	B1: nonstricturing, nonpenetration	522 (55.8%)
	B2: stricture	201 (21.4%)
	B3: fistulizing	104 (11.1%)
Gender	Male	482 (51.6%)
	Female	453 (48.4%)

**Table 5 medicina-55-00704-t005:** Univariate analysis of different factors involved in the presence of the predominant phenotype.

	Univariate Analysis (*p*-Value)							
	Ulcerative Colitis			Crohn’s Disease				UC=CD
Characteristics	NV	C	NE	W	S	SE	B	SV
Age (year)	0.12	0.64	0.007	0.06	0.004	0.47	0.05	0.001
Gender (male)	0.99	0.88	<0.001	0.60	0.01	0.05	0.29	0.63
Family history	<0.001	<0.001	0.65	0.08	0.13	0.74	0.58	0.23
Smoking status	0.99	0.57	0.99	0.03	0.35	0.54	0.007	0.60
Provenience								
Rural	0.02	0.83	0.04	0.45	0.36	0.53	0.6	0.75
Urban	0.64	0.85	0.86	0.33	0.36	0.78	0.89	0.75

B = Bucharest, C = Center, NE = North East, NW = North West, SE = South East, S = South, SW = South West, W = West; (UC=CD) = the proportion of patients with ulcerative colitis is the same with the proportion of patients with Crohn’s disease.

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
