# Peer review of "Geographic Distribution, Phenotype and Epidemiological Tendency in Inflammatory Bowel Disease Patients in Romania"

_medicina, 2019, doi:10.3390/medicina55100704_

Round 1

Reviewer 1 Report

The present article considers a quite long period to investigate IBD epidemiology in Romania. The prospective design of the study allows to define the characteristics, the trends and the phenotype of IBD in the selected population. Authors chose an interesting topic and provided a useful contribution for clinicians with their results from a large database. Statistical Analysis are properly performed and results are in line with the aim of the study. The lack of information regarding private centers for IBD is unfortunately a weakness for the study.

Some minor revisions have to be addressed as follows:

In the abstract please write Inflammatory bowel disease the first time and then IBD. Page 10 line 212 and 214 and Conclusion paragraph: please correct ‘’Crohn disease’’ with ‘’CD’’ Page 10 line 212 and Conclusion paragraph: please correct ‘’ulcerative colitis’’ with ‘’UC’’ Figure 1 shows a poor quality and could be improved Figure 2 needs a legend in order to be self-explaining Table1, Table 2 and Table 5 need a legend in order to be self-explaining

Author Response

Dear reviewer,

Thank you for your response and for your suggestions. We have corrected in the manuscript with the abbreviations for inflammatory bowel disease, ulcerative colitis and Crohn disease accordantly to your suggestions.

For “Figure 1” we change the figure with a new one, with better quality and now we put the English name of the regions on it, also we put the legends on Figure 2, Table 1, Table 2 and Table 5. Thank you again for notice these mistakes.

Regarding the lack of information of private centers, we already write this in the manuscript that is not entirely a population-based study, because not all the medical doctors contributed to this study and further studies are needed, but we hope that in the future we can do something bigger, with all IBD patients involved.

King regards.

Reviewer 2 Report

Goldis and colleagues performed an interessant work about epidemiology of IBD in a east european country. Data were presented in a clear and accurated manner. Only the figure 1 can be modified with the english name of regions.

Author Response

Thank you for your review and for the sugestions. We have modified the manuscript accordantly. We change Figure 1 with another figure with better quality and with the regions in english.
